# Medieval Monasticism in Iceland and Norse Greenland

Steinunn Kristjánsdóttir 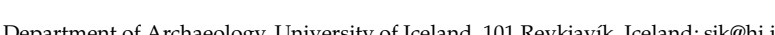

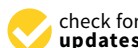



Department of Archaeology, University of Iceland, 101 Reykjavík, Iceland; sjk@hi.is

**Abstract:** The aim of this paper is to provide an overview of the monastic houses operated on the northernmost periphery of Roman Catholic Europe during the Middle Ages. The intention is to debunk the long-held theory of Iceland and Norse Greenland's supposed isolation from the rest of the world, as it is clear that medieval monasticism reached both of these societies, just as it reached their counterparts elsewhere in the North Atlantic. During the Middle Ages, fourteen monastic houses were opened in Iceland and two in Norse Greenland, all following the Benedictine or Augustinian Orders.

**Keywords:** Iceland; Norse Greenland; monasticism; Benedictine Order; Augustine Order

The aim of this article is to provide an overview of the medieval monastic houses operating in the northernmost dioceses of the Roman Catholic Church: Iceland and Norse Greenland. At the same time, it questions the supposed isolation of these societies from the rest of the Continent. Research on activities in Iceland and Greenland shows that the transnational movement of monasticism reached these two countries as it reached other parts of Northern Europe. Fourteen monastic houses were established in Iceland and two in Norse Greenland during the Middle Ages. Two of the monastic houses in Iceland and one in Norse Greenland were nunneries, whereas the others were monasteries. Five of the monasteries established in Iceland were short lived, while the other nine operated for centuries. All were closed due to the Reformation around the mid-sixteenth century. On the other hand, the monastic houses in Greenland were closed around the time the Norse settlement there vanished in the fifteenth century. The monastic houses in both Iceland and Norse Greenland belonged to either the Benedictine or Augustinian Orders.

## 1. The Background of Icelanders and the Norse Greenland Settlers

Iceland was initially settled by immigrants who came mainly from western Scandinavia, the British Isles, and Ireland during the last decades of the ninth century. This has been confirmed by genetic studies, which demonstrate that the early Icelanders were a combination of Norse, Gaelic, and other mixed heritage (Ebenesersdóttir et al. 2018). Furthermore, recent studies show that the settlement period was a long one, as people continued to migrate to Iceland at least until after the formal conversion to Christianity in 999 or 1000 (Vésteinsson and Gestsdóttir 2016). The voyages of that time continued, but two areas of southwestern Greenland were settled by Icelandic settlers around the turn of the tenth century. Shortly thereafter, Icelanders extended their expeditions to North America (Eiríks Saga Rauða 1953, pp. 326–29; Karlsson 2000, pp. 28–32; Guðmundsson 2005, pp. 11–19).

The two areas settled in Norse Greenland were the Eastern Settlement, which was settled first, and the Western Settlement. Greenland had indeed been inhabited by several indigenous cultures long before the arrival of the Icelandic settlers. However, none of the indigenous cultures lived in southwestern Greenland at the time of the Norse settlement. Still, by 1500, the Thule culture occupied most coastal areas of Greenland, including the Eastern and Western Settlements, where Icelandic settlers resided from approximately 1000 to 1450 (Gulløv et al. 2004, pp. 11–24). Genetic studies on skeletal remains from Norse Greenland do not, however, show any indications of mixture between the people of the Thule culture and the Norse (Lynnerup 1998, pp. 34–38, 120–28).

The cultural background of those settling in Iceland from 870 to 1000 is observed as having been a combination of different religious views and habits originating from both Old Pagan and Christian practices, but the Christianisation of the societies in Northwest Europe had been ongoing for a while when Iceland was settled. The earliest Christian cemeteries excavated in Iceland date to the turn of the eleventh century, yet quite a few of them include burials that are older than the conversion itself, indicating that Icelanders had soon become overtly acquainted with Christian burial customs (Hugason 2000, pp. 18–30, 66–73; Vésteinsson 2005, pp. 72–76; Vésteinsson et al. 2019, pp. 171–77). Archaeological research undertaken in Norse Greenland accords with the developments described above, implying further that the first settlers were already Christian upon their arrival. No pre-Christian graves have been found with any certainty in Norse Greenland (Keller 1989, pp. 51–109, 210–12; Lynnerup 1998, pp. 9–10, 51; Arneborg et al. 1999, pp. 159–66; Arneborg et al. 2012, pp. 2–37).

Soon after the settlement of Iceland, the settlers formed their own system of government based on chieftaincies. There was no central government, but district assemblies were held regularly throughout the country. The main assembly, *Alþingi*, was established in 930 in Þingvellir in southern Iceland, where the district chiefs, *goðar*, gathered to introduce new legislation or to settle major disputes that concerned the whole nation. The *goðar* became the most powerful men in the country, along with the president of *Alþingi*, called the lawspeaker. Written descriptions indicate (see, for example, (Flateyjarbók 1945, pp. 231–32)) that the Icelanders adapted this governmental system of chieftaincy similar to the society in Norse Greenland. It is worth noting, however, that the secular governance of both Iceland and Norse Greenland was taken over by the Norwegian crown in 1262. From then until approximately 1381, executive power rested with the Norwegian king and his local officials (Karlsson 2000, pp. 83–86, 89–95). At that time, the population of Iceland is estimated at 40,000 (Karlsson 2000, p. 45) and the population of Norse Greenland at 2500 to 3000 (Guðmundsson 2005, p. 17).

Consequently, the society in Norse Greenland appears to have been organised in the same manner as in Iceland (see even (Krogh 1982a, pp. 9–26; Gulløv et al. 2004, pp. 219–40)). However, Norse Greenlandic society may have been more 'ecclesiastically' organised than Icelandic society, as it was composed entirely of Roman Catholic Christians. Recent studies show, for example, that from the beginning, Greenlandic parishes were larger and far more centralised than Icelandic ones (Vésteinsson 2010, pp. 140–49). Nevertheless, as in other Christian societies of Northwest Europe during the early Middle Ages, the Church in Iceland and Norse Greenland was based on the proprietary system.[1] A proprietary church was an ecclesiastical house founded by a landowner on his own property, and where he maintained rights of investiture. In other respects, the proprietary churches were meant to serve a larger community, usually a parish, although some of the early churches were built as private chapels intended for the sole use of a single family.[2] However, the establishment of the churches, both private and proprietary, was generally based on the private initiative of farmers, chieftains, and even bishops (Smedberg 1973, pp. 88–99; Keller 1989, pp. 212–14; Wood 2013).

During the eleventh century, the establishment of proprietary churches reached a peak in Europe, including in Iceland, causing a serious conflict between the secular and ecclesiastical authorities on the Continent. The conflicts were launched by Pope Gregory VII, who, during his tenure from 1073 to 1085, strived to differentiate administratively between secular and ecclesiastical powers. His reforms were seen, however, as bringing about a significant undercutting of secular power, including prohibiting laymen from interfering in ecclesiastical matters—in particular, the operation of churches (Wood 2013, pp. 850–64). The conflicts that arose, termed the Investiture Controversy, were more or less settled on the mainland in 1122, but had not yet started in Iceland. It was not until after the establishment of the archdiocese in Niðarós (Trondheim, Norway) in 1153 that the archbishops there started to claim rights over the churches in their provinces, as had occurred on the Continent, in addition to acquiring the right to appoint churchly officials

such as bishops and abbots (Arnórsdóttir 1995, p. 108; Guðmundsson 2000, pp. 18–24). The conflicts in Iceland, called *Staðamál*, lasted much longer than those on the Continent, pitting the *goðar* against religious chiefs such as bishops Þorlákur Þór-hallsson (1178–1193) and Árni Þorláksson (1269–1298), who supported the ongoing reform of the Church in the country. Because Iceland had been under Norwegian control since 1262, the conflicts were finally settled with an official agreement made at Avaldsnes in Norway in 1297, albeit after negotiations between royal and ecclesiastical authorities. The agreement emphasises that the operation of churches and the appointment of bishops, abbots, abbesses, and priors—specifically, in Iceland, Greenland, the Faeroe Islands, and the Hebrides—should be undertaken in consultancy with the archbishop in Niðarós (DI II 1893, pp. 325–28). The proprietary church system was not completely abandoned in Iceland, however, as some of the churches there remained in private hands throughout the Middle Ages (Stefánsson 2002, pp. 155–64; Karlsson 2000, pp. 96–99).

The growth of the proprietary church system in Iceland and the initiative to build private chapels can be seen in an increased number of farms with churches dating to the first two centuries after Christianisation. In Greenland as well, single churches seem to have grown quickly in number soon after the settlement there, as church ruins have thus far been identified in seventeen places in the Eastern Settlement and two in the Western Settlement (Smedberg 1973, pp. 89–90; Keller 1989, pp. 212–14, 262–65; Vésteinsson 2010, pp. 139–40). At the same time, monasticism gained a firm foothold in both Iceland and Norse Greenland, with fourteen monastic houses founded in Iceland and two in Norse Greenland.

## 2. The Expansion of the Roman Catholic Church

Icelanders decided to formally adopt Christianity as their official religion in 999/1000, and the Norse societies of the Orkney Islands, Faeroe Islands, and Norway officially converted around the same time. Ireland, Scotland, Wales, England, and Germany had all adopted Christianity some centuries earlier, and Denmark had followed in approximately 965 (Sigurðsson 2008, pp. 66–77; Walaker Nordeide 2011, pp. 79–83). Moreover, as early as 1022, Pope Benedict VIII (r. 1012–1024) declared that Denmark, Norway, Sweden, and Iceland should belong ecclesiastically to the archbishopric of Hamburg (DI I 1857–1876, pp. 51–53). Pope Leo IX (r. 1049–1054) reinforced this in 1053 when he reinstated Archbishop Adelbert (r. 1043–1072) in office, but in his declaration that year, he listed Greenland for the first time as one of the countries belonging to the archbishopric of Hamburg (DI I 1857–1876, pp. 57–60).

The Roman Catholic Church had certainly strengthened its position in Iceland by the time that Icelanders received their first bishopric in Skálholt in 1056. The expansion of the Church did not become evident, however, until after Pope Paschal II (r. 1099–1118) had transferred the Nordic countries from the archdiocese of Hamburg in 1104 and placed them under a separate archdiocese in Lund. Pope Paschal II also appointed the incumbent bishop of Lund, Asser Thorkilsen (r. 1104–1137), to serve as Lund's first archbishop, whereupon Asser continued to strengthen the position of the Church in its northernmost dioceses of Europe. Two years later, in 1106, Archbishop Asser established the bishopric of Hólar in Iceland. Shortly thereafter, a bishopric was founded in Kirkjubær in the Faeroe Islands, and finally, in 1124, a bishopric was established in Garðar in Greenland. Three years earlier, Asser had appointed an Icelander, Eiríkur Gnúpsson, as bishop of Greenland and the Norse settlements in North America. Around that time, the first bishop of Hólar, Jón Ögmundsson, laid the groundwork for the earliest successful Benedictine settlement in Iceland, Þingeyraklaustur, which began operating in 1133 (Jensson 2016, pp. 20–22). Yet, when the archdiocese of Lund was split up in 1153 and the bishoprics in Iceland and Norse Greenland became part of the archdiocese of Niðarós, along with Norway, the Faeroes, the Northern Isles, the Hebrides, and the Isle of Man—all with their own bishoprics—the activities of the Church in Northern Europe expanded immensely (Figure 1). Its growth could be seen not at least in new monastic orders and monastic houses from the late

eleventh and twelfth centuries (Aston 2001, pp. 9–10). In Denmark and Norway, new monastic foundations increased greatly in number during the twelfth century, most of them following the Benedictine and Augustinian Orders, although there were Premonstratensian, Franciscan, Dominican, and Cistercian establishments as well. In Denmark, forty-nine of the approximately 140 monastic houses ever founded in the country were established during the twelfth century, as were seventeen of the twenty-seven monastic houses founded in Norway. The establishment of new monastic houses stalled again in both countries during the second half of the thirteenth century (Gunnes 1987, pp. 51–66; Lidén 1993, p. 65; Olsen 1996, p. 24; Jakobsen 2005).

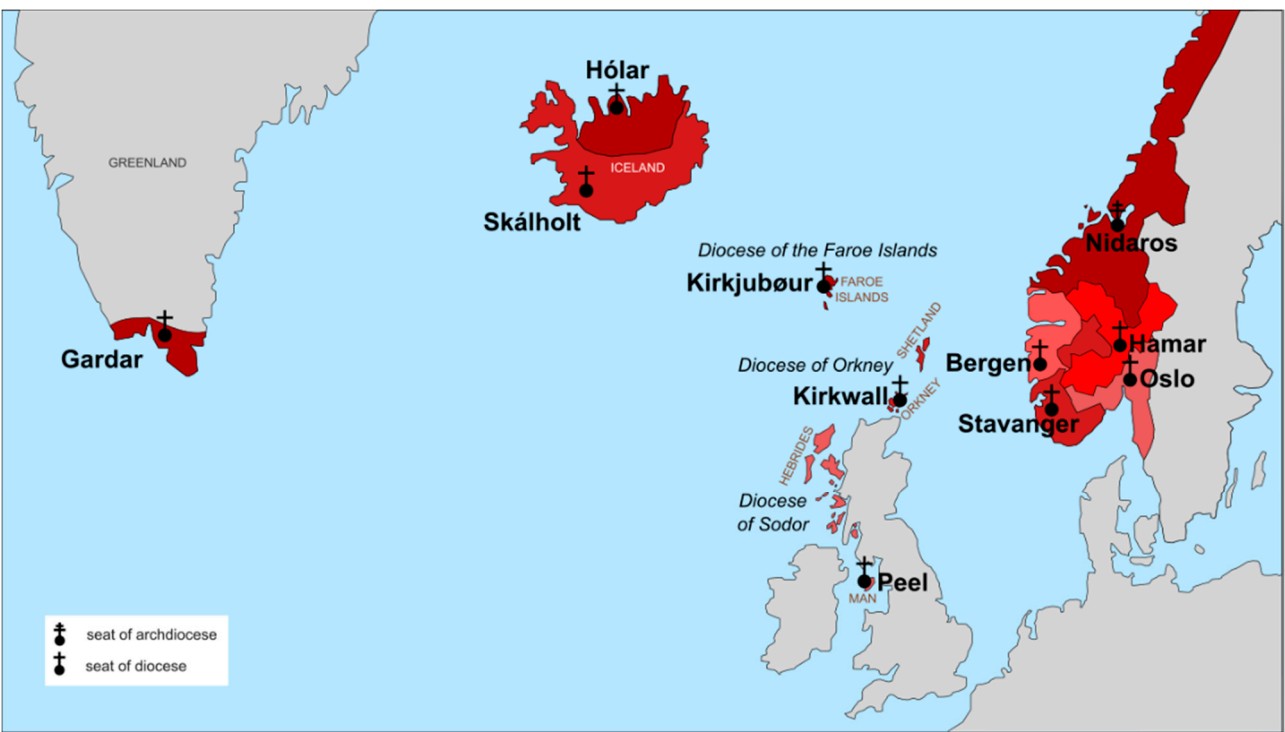

**Figure 1.** The ecclesiastical province of Niðarós 1153–1387.

In Iceland, three new Benedictine monasteries followed soon after the founding of Þingeyraklaustur: Munkaþverárklaustur (1155) and Hítardalsklaustur (1166) in Hólar bishopric, and Iceland's first nunnery, Kirkjubæjarklaustur (1186), in Skálholt bishopric (Kristjánsdóttir 2017). In Greenland, the Benedictine nunnery located in Uunartoq fjord may also have been founded during this initial phase of Benedictine expansion to the northernmost territories of Europe (Clark 2011, p. 53; Jensson 2016, pp. 20–24). The Augustinians began circulating their agenda as well, establishing the monasteries of Þykkvabæjarklaustur (1168) and Flateyjarklaustur/Helgafellsklaustur (1172/1184) in Skálholt bishopric in Iceland (Kristjánsdóttir 2017). The monastery in Tasermiut fjord in Greenland was probably established by then as well (Vebæk 1953, pp. 195–200).

Nevertheless, the earliest attempt to establish a monastic institution in Iceland took place before Roman Catholic Christianity had gained such a firm foothold in Northern Europe. This was Bæjarklaustur, which was established in 1030 but discontinued operation after only two decades. Bæjarklaustur was founded by a Benedictine monk, Rúðólfur, who was sent to Iceland in order to establish the monastery on behalf of the archbishop of Hamburg (Kristjánsdóttir 2017, pp. 67–78). A similar attempt to establish a monastic house appears to have been made in Nidarholm in Norway in 1028, but it failed as well (Walaker Nordeide 2011, p. 117; Haug 2014, p. 206). The early Benedictine monastery established in Schleswig (by then within the kingdom of Denmark) in 1025 continued to operate for nearly two centuries, however (Nyberg 2000; Jakobsen 2005, p. 58).

In addition to the monastic houses founded in Iceland during the twelfth century, three were established on the initiative of wealthy chieftains between 1193 and 1226, when the *Staðamál* were still unsettled. These monasteries were Keldnaklaustur, Saurbæjarklaustur, and Viðeyjarklaustur, and their founders became their lay abbots, as was customary on the mainland (Kristjánsdóttir 2017). Furthermore, while the *Staðamál* were at their peak in Iceland, all of the active monasteries and the only nunnery in operation at that time were confronted with defensive struggles that some of them—such as Hítardalsklaustur, Keldnaklaustur, and Saurbæjarklaustur—did not survive. Soon after the *Staðamál* were settled, the Church became particularly powerful in Iceland, both governmentally and economically. Two new monastic houses were founded right around the time the disputes were resolved: the Benedictine nunnery Reynistaðarklaustur (1295) and the Augustinian monastery Möðruvallaklaustur (1296), both belonging to Hólar bishopric. The growth of the Church continued until the Reformation, but the last monastery in Iceland was established in Skriðuklaustur in 1493 (Kristjánsdóttir 2017, pp. 46–49), by which time the Norse settlement in Greenland had vanished.

Due to a lack of sources, it is not known precisely when the Benedictines and Augustinians arrived in Norse Greenland. It may have been during the initial expansion phase of monasticism to Northern Europe as described above, or later, when the Roman Catholic Church had gained a proper foothold there by end of the thirteenth century with the resolution of the *Staðamál*. On the other hand, written sources on the monastic houses run in Iceland are abundant. Their foundation dates are known in nearly all cases, as are their orders, the assessed incomes of the larger ones, their furnishings, their internal activities and noteworthy events, the dates of their dissolution, and the names of their chiefs (abbots, abbesses, priors, and prioresses). Furthermore, the names of other residents such as corrodians, novices, male and female students, and lay workers are known in many cases. Knowledge of the existence of the Norse Greenland monastic houses is based almost entirely on a description made by Ívar Bárðarson, an official agent of the bishopric of Bergen who went to Greenland to record the churches located there in the mid-fourteenth century (see, for example, Halldórsson 1978, pp. 133–37). In his records, Ívar lists the two monastic houses, a Benedictine nunnery and an Augustinian monastery, as well as eight churches. Both appear to be rich in land, but like the successful ones run in Iceland, their estates must have been the mainstay of their economic strength. The nunnery is reported as owning nearly all of the farms in the fjord where it was located and half of the islets there, while the bishopric in Garðar owned the other half. The monastery is described as 'large', but no further detail is given. The record states that it was dedicated to St Olav the king and owned all farms in the fjord where it was based (see, for example, Halldórsson 1978, p. 135).[3]

The monastic houses in Norse Greenland are also briefly mentioned 22 June 1308 in a letter from Bishop Árni Sigurdsson in Bergen designated to Bishop Þórður in Garðar, thus providing further proof of their existence. In the letter, Bishop Árni expresses his gratitude to Bishop Þórður by sending him various gifts for having prayed for the soul of King Erik II (d. 1299), as well as the souls of five recently deceased Norwegian bishops. A portion of the gift was intended for the monastic houses in Greenland, including skins, a light blue *chaperon,* and a gown made of the same fabric. Also given was a barrel of grapes (Grønlands Historiske Mindesmærker 1845, pp. 94–98[4]). Interestingly, on the same day, Bishop Árni in Bergen sent almost an identical letter to Bishop Árni Helgason in Skálholt in gratitude for him having prayed for the soul of King Erik II and five recently deceased Norwegian bishops. Gifts to the bishop of Skálholt included, on the other hand, wax and beer, but no grapes or textiles (DI II 1893, pp. 362–63).

## 3. Research on the Monastic Houses in Iceland

Monastic and Church archaeology has been growing as a field of research for some time in most Northwest European countries (McClain 2012, pp. 131–70; Gilchrist 2014, pp. 235–90). Iceland has been part of this trend. Four monastic sites—Viðeyjarklaustur,

Kirkjubæjarklaustur, Skriðuklaustur, and Þingeyraklaustur—have been fully or partly excavated in recent years. In addition, all fourteen monastic sites in Iceland were recently surveyed and documents on their activities systematically investigated (Kristjánsdóttir 2017).

The excavations carried out to date on the four Icelandic monastic sites investigated archaeologically vary in scale and scope. The excavation on the island of Viðey, where the Augustinian monastery Viðeyjarklaustur was located, began in 1987 due to construction work and continued until 1995. During this period, the ruins of the monastic house there were not identified with any certainty, perhaps due to the long history of occupation on the spot where the monastery is supposed to have stood. Viðeyjarklaustur is also the only monastic house to be demolished in an attack by Protestant followers during the initial phase of the Reformation in Iceland. This raid, which took place in 1539, forced the closure of the monastery after over three centuries of successful operation. After that, the monastic buildings were converted to a residency for the superintendents of the Danish king, who were tasked with caring for the properties formerly owned by the monastery. However, some artifacts from the monastic period were found, including a likeness of St Dorotha and three wax tablets (Hallgrímsdóttir 1991, pp. 102–21; Kristjánsdóttir 1995; Kristjánsdóttir 2017, pp. 325–55). Like many other monastic houses, Viðeyjarklaustur became very wealthy after the *Staðamál* had been settled. In 1313, it owned fourteen farms, but by 1395, the number of farms it owned had risen to fifty (DI II 1893, p. 377; DI III 1896, pp. 597–98).

In contrast to the excavation in Viðey, the excavations conducted on the sites of the Benedictine nunnery in Kirkjubær and the Augustinian monastery at Skriðuklaustur were organised as research projects. Both excavations started in 2002, and both were also intended from the outset to locate and investigate the lodgings and activities of monastic houses on the two sites. Kirkjubæjarklaustur was run for nearly four centuries, from 1186 to 1541, albeit with a break of approximately seven decades during the thirteenth century due to the *Staðamál* conflicts (Kristjánsdóttir 2017, pp. 274–80). Skriðuklaustur, on the other hand, was established in 1493 and had only been in operation for five decades when the clergy of Skálholt bishopric officially changed their allegiance from Catholicism to Lutheranism in 1541. The residents were allowed to remain in their institutions afterwards, however, as was permitted in the other administrative provinces of the Danish kingdom, to which Iceland belonged by then. The residents in Kirkjubæjarklaustur had all left the nunnery in 1548, and Skriðuklaustur residents left in 1553 or 1554 (Kristjánsdóttir 2012; Kristjánsdóttir 2017, pp. 271–97, 419–49). Iceland remained semi-Catholic and semi-Lutheran from 1541 to 1550 because the bishop of Hólar diocese, Jón Arason, refused to accept the Lutheran Church ordinance and remained the representative of Catholics until he was forcibly removed from his post and executed late in 1550. The remaining monastic houses—three monasteries and one nunnery belonging to Hólar bishopric—were dissolved the following year (Kristjánsdóttir 2017).

During the archaeological investigation in Kirkjubæjarklaustur, which lasted for five years, only the northeastern corner of the nunnery's lodgings was excavated. The parts excavated were nevertheless sufficient to show that the nunnery was built with the rooms arranged in a square by a cloister garth. No graves were exhumed, however. Pieces of glass decorated with ecclesiastical images were found, as was an altar stone. No other ecclesiastical items were found (Mímisson and Einarsson 2009, pp. 44–49). Even so, the findings supported what may be read from written sources that place primary importance on the nunnery's textile work, which was officially praised by Bishop Vilchin (r. 1391–1405) during a visitation in 1397. Soon after his visitation, Bishop Vilchin purchased tapestries from the nuns to cover all four walls of the so-called large hall in Skálholt episcopal see, as well as a number of ecclesiastical garments for the cathedral (Lögmannsannáll 1888, pp. 287–88). Moreover, an inventory made in Kirkjubæjarklaustur in 1343 listed not only wall hangings around the nunnery's church but also fourteen antependia, twelve cloths for the lectern, twenty-two chasubles (including twelve made of elaborate silk), twenty gowns, six dalmatics, and eight robes. The most elaborate chasuble, which was blue, is listed separately (DI II 1893, p. 781; DI IV 1897, p. 238; DI VIII 1906–1913, p. 5). This

means that in Kirkjubæjarklaustur there were enough chasubles for at least twenty priests in 1343. In 1397 there were twenty-four chasubles. Given that there never seem to have been more than one or two priests serving the nuns in Kirkjubæjarklaustur at any time, the extra chasubles may have been stock from their textile manufacturing. Embroidered chasubles, gowns, antependia, and tapestries were extremely valuable, as were most other ecclesiastical textiles. An elaborate chasuble, for example, could have a value comparable to that of three or four farms (Jónsson 1915–1929, p. 234). In fact, it even is safe to say that ecclesiastical tapestries were even more desirable among the clergy and the aristocracy than most other works of art and crafts, including books, during the Middle Ages. Although the excavation of the ruins of Kirkjubæjarklaustur covered only one corner of the nunnery's lodgings, it revealed clear evidence of textile work, besides miscellaneous utensils for the household. In the ruins, the remains of a nearly complete warp-weighted loom were found, in addition to at least 48 single loom weights, ten needles, and four spindle whorls (Mímisson and Einarsson 2009, p. 49; Parsons 2018). It is also noteworthy that the nunnery had a significantly larger number of sheep than any other monastic house in Iceland when the livestock inventory was compiled in 1343. By then, the home farm of Kirkjubæjarklaustur had ninety-four cattle and nearly 750 sheep, which were apparently bred to produce wool for the nunnery's textile production (DI II 1893, p. 781; DI VIII 1906–1913, p. 5). In contrast, the other monasteries ran large cattle ranches, most likely for manufacturing manuscripts. According to an inventory made due to its temporary closure in 1218, Kirkjubæjarklaustur already owned all or part of nine farms by then, in addition to receiving rents from another nine farms owned by others (DI I 1857–1876, p. 394). Farms owned by Kirkjubæjarklaustur were inventoried two more times, and both inventories show considerable growth. In 1343, the nunnery owned nineteen farms, and by 1397, it owned twenty-three (DI II 1893, p. 781; DI IV 1897, pp. 238–40).

The excavation in Skriðuklaustur, which began in 2002, continued until 2012. During this ten-year period of research, the monastic ruins, including the adjacent cemetery, were excavated in their entirety. The monastic complex and the adjoining church, both partially two-storied, occupied an area of just over 1500 square metres. Thirteen rooms were detected, used for both ecclesiastical and secular work (Kristjánsdóttir 2012, pp. 59–64). A total of 298 graves were discovered in the monastic cemetery, some inside the church and in the rooms in close proximity to it. All of the burials provided, through their presence, important information about the residents of the monastery. The bones of foetuses, neonates, young children, adolescents, and adults (both men and women) were discovered in the graves, and more than half of the skeletons showed identifiable signs of various chronic diseases, illnesses, or traumas. Besides this, both the artefacts and non-native medicinal plants found on the site reveal that Skriðuklaustur had served as a hospital during its operation, as many other monasteries on the mainland did. It is worth noting here that Skriðuklaustur never served as a parish church because the church serving the people living in the valley was situated only two kilometres away (Kristjánsdóttir 2012, p. 277–83). Further examination of the human bone collection from Skriðuklaustur showed that imported mercury was also used as a treatment for healing diseases such as syphilis, and indeed, at least sixteen individuals with syphilis were buried in the monastic cemetery there (Kristjánsdóttir 2011; Kristjánsdóttir 2012, pp. 198–207; Walser et al. 2019, pp. 48–61). In addition to the mercury, imported healing plants, surgical equipment, and the bones of seabirds and sharks caught in the waters off Iceland's eastern coast were found at the site, as were fragments of pottery and an abbey token from the south of France. A likeness made in the Netherlands of St Barbara, one of the fourteen saints venerated as holy helpers, stood in the church. A horn for calling the canons to services was also imported from the Netherlands; it was found broken in the monastic ruins, as was the figurine of St Barbara. Skriðuklaustur owned forty-one farms at the time of its dissolution in 1541 (Kristjánsdóttir 2012; Kristjánsdóttir et al. 2014).

The principal aim of the ongoing research at Þingeyraklaustur is to investigate the manuscript making that took place there. The research, therefore, has roots in literary

studies and archaeological excavation (Kristjánsdóttir 2018). Þingeyraklaustur operated uninterrupted from 1133 to 1551 and thus became the most successful and longest lived of all Icelandic monastic houses. It soon gained a prestigious reputation for its manuscript and book production—a reputation that reached far beyond the country's borders (see the article by Jensson in this volume). Not only were the Benedictine monks there involved in writing religious and historical texts in Latin and Old Norse/Icelandic for Christian kings in northern Europe, but they were also responsible for writing histories for some of the leading chiefs—ecclesiastical and secular—in Iceland (see also Jensson 2017, pp. 875–49).

The surveying of all fourteen monastic sites in Iceland, performed during the period from 2013 to 2017, proved that the nine successful houses became well staffed with both religious and lay members, as the excavation of Skriðuklaustur had shown. The sources showed, moreover, that these nine monastic houses came to be wealthy landholders that owned a number of valuable farms, particularly after the *Staðamál* conflicts ended in the late thirteenth century. It is estimated that they owned approximately 10% of the 5000 farms in the country, or an average of fifty farms per monastery. In 1525, for example, Þingeyraklaustur owned 100 farms and Munkaþverárklaustur sixty-three (DI IX 1909–1913, pp. 309–16). The farms were either donated to the monastic houses for religious reasons or purchased by their chiefs, abbots, or abbesses. In addition, monasteries and nunneries alike successfully ran large cattle and sheep ranches, manufactured manuscripts and textiles of various kinds, and provided a broad range of social services to their local communities. Most of them made agreements with corrodians but also hired lay workers to carry out the work that was needed. The monastic houses even provided novitiate placements for future canons and nuns, as well as offering academic and vocational training to boys and girls, as they did widely in medieval Europe. Simultaneously, the monastic houses had a broad range of mandatory tasks to carry out in the name of charity and salvation (Kristjánsdóttir 2017; see also the article by Åsen in this volume).

## 4. Research on the Monastic Houses in Norse Greenland

There is no reason to doubt that two monastic houses were established in Norse Greenland, although sources about their founding and activities are very sparse. Both were situated in the Eastern Settlement, but what has made the exact locations of these two monastic sites uncertain is the inaccurate placename usage in Ívar Bárðarson's descriptions. Ívar says that the monastery is located in Ketilsfjörður and the nunnery in the next fjord, which he calls Hrafnsfjörður (Halldórsson 1978, p. 135; Guðmundsson 2005, p. 24). As we will see, the location of the ruins of the Benedictine nunnery may have been confirmed during an excavation conducted in Uunartoq fjord in 1945–1946, but the supposed ruins of the Augustinian monastery have only been surveyed, so their location is still uncertain (Roussell 1941, pp. 48–51; Lynnerup 1998, pp. 20–22, 30). Nevertheless, the estates of the monastic houses in Norse Greenland indicate their strong position in the society there. The nunnery is estimated to have owned thirty to thirty-five farms and the monastery ten to fifteen farms, based on the number of farms identified in their immediate vicinity (Vésteinsson 2010, p. 146).

During the expedition to the supposed site of the nunnery, named Narsarsuaq (labelled Ø149) in Uunartoq fjord (Hrafnsfjörður), twenty-five ruins were detected in 1945–1946. The ruin of a church and a surrounding cemetery was excavated, along with six ruins belonging to the site complex (Vebæk 1991, pp. 27–28). Skeletons of at least fifty-seven individuals were exhumed, and an unspecified number of loose bones were detected scattered around the site, in addition to one mass grave. Not all of the bones were collected, and only the upper layers of burials in the cemetery were excavated. Still, twenty of these fifty-seven were found inside the church, while the rest were found in the surrounding cemetery (Vebæk 1991, pp. 28–43, 44–46). Some of the human bones were radiocarbon dated to the period 1322–1428 (Lynnerup 1998, pp. 24, 147–48; Arneborg et al. 1999, p. 161). This dating fits well with the time of the Norse occupation in the Eastern Settlement. Vague traces of an earlier church or building were detected beneath the church ruin, indicating that

the site covered two phases of the Norse settlement: one from approximately 1000 and the second one from the time of the nunnery there (Vebæk 1991, pp. 25, 27–28; Lynnerup 1998, pp. 20–22). What surprised Vebæk, however, was that people of both sexes and all ages were exhumed from a cemetery belonging to a nunnery. Nine of those buried there were children up to age 12/14, two were subadults aged 12/14 to 18/21, thirty were adults up to age 35, eight were of mature age, and another eight were of unknown age at death. There were twelve females and five males, but in forty instances the person's sex could not be identified, indicating that the site may have been occupied mainly by females (Vebæk 1991, pp. 31, 46; Lynnerup 1998, p. 45). Vebæk (1991, p. 31), therefore, suggested that the nunnery's church may have been a parish church, in addition to serving the nuns. This could certainly have been the case, but the fact is that nunneries were dependent on stewards and priests for both household labour and religious services, an arrangement that often led to the combination of nunneries with parish churches in one or another way (Gilchrist 1994, p. 90). The recent investigations on the Benedictine nunneries run in Iceland, at Kirkjubæjarklaustur and Reynistaðarklaustur, show that they were inhabited mainly by females, although there were also other residents of both sexes and all ages. Among these were male stewards, the nunnery's priest, and male and female lay workers who cared for the nuns' livestock. The stewards and lay workers even had their families residing with them in the nunneries. Other residents included novices and students of both sexes, as well as corrodians who brought their families and servants with them to their corrody (Kristjánsdóttir 2017). The cemetery in Narsarsuaq may, therefore, represent a community similar to those found in the Icelandic nunneries.

Vebæk was also concerned about the lack of findings proving that the ruins of the settlement complex in Narsarsuaq were actually of the nunnery, as fragments of a church bell were the only ecclesiastical item found (Vebæk 1991, pp. 74, 75). However, browsing through the list of findings from the excavation there strongly suggests activities like those expected in a nunnery: the miscellaneous utensils for a common but large household, besides the various tools and equipment needed for textile work (see Vebæk 1991, pp. 75–80). The large number of artifacts connected to the household, such as barrels, vessels, plates, spoons, and ladles, imply abundant food supplies for the houses in Narsarsuaq, as can in fact be observed from inventories of the Icelandic nunneries. Moreover, research on nunneries in England shows that the nunneries there normally produced surplus foodstuffs, which enabled them to feed their own residents, provide food charity to the wider community, and even sell it for profit, as they did with their textile production (Gilchrist 1994, pp. 85–88). The remains of the household and the site complex—counting twenty-five ruins—certainly show that this was a residence housing many people, but spindle whorls, weight looms, and needles found during the excavation are evidence of work similar to that performed in Kirkjubæjarklaustur in Iceland. It is nevertheless worth noting that spindle whorls are among the most frequently found tool in excavations of living quarters in both Iceland and Norse Greenland, as they were necessary for making clothing of any kind.

The supposed site of the Augustinian monastery at Taserminutsiaq (labelled Ø105) in Tasermiut fjord (Ketilsfjörður) was located in 1932 (Vebæk 1991, pp. 5–6). No excavations have taken place on the settlement and only eight ruins have been detected on the surface ground there. Among them is a ruin of a church and cemetery (Roussell 1941, pp. 48–51; Vebæk 1953, pp. 195–196; Krogh 1982b, p. 286). The Augustinian monastery Skriðuklaustur, which is the only monastic house that has been excavated in its entirety in Iceland—and in fact the northernmost one to have been excavated in Europe—in fact differs vastly in size and organisation from the site in Taserminutsiaq. Ívar Bárðarson had claimed that the monastery there was 'large', but the ruins in Taserminutsiaq do not indicate a large-sized house complex. Only an excavation could verify this.

## 5. Conclusions

Research undertaken on the activities of the monastic houses in Iceland and the Benedictine nunnery in Norse Greenland indicate that their operation was in line with practices in other monastic houses of the Benedictine and Augustinian Orders in Europe. By and large, they served not only the Church itself but also the needs of the surrounding society. In addition to fulfilling their mandate of providing charity and salvation, the monastic houses produced their own agricultural products, textiles, and books, and they even offered academic education and vocational training based on the fields in which they specialised. Some monastic houses even offered hospital services, as can be seen in the case of Skriðuklaustur.

Generally, the business of the monastic houses was based on donations or alms, in addition to rental income and revenues from their own manufacturing. In this way, the monastic estates were extremely important for the agricultural revenue that they generated, and coastal farms provided the monastic houses fishing rights and access to stranded valuables on their beaches. Moreover, the multiplex production of the monastic houses required a range of raw materials, either imported or locally provided through the herding of domestic animals on their farms. At the same time, a large lay workforce was needed to collect and utilise the diverse valuables, and religious personnel with appropriate academic capacity were necessary to carry out the ecclesiastical work. Similarly, running the monastic household demanded considerable labour, with the household work and herding usually carried out by the laity and the mandatory work of charity and salvation in the hands of the religious personnel. This may at least be observed from the various sources on the monastic houses in Iceland and even from the supposed ruins of the Benedictine nunnery in Norse Greenland. Written and archaeological sources prove that this pattern characterised life in the monastic houses in Iceland, and there is no reason to believe that those located in Norse Greenland were run differently. Moreover, the nunnery in Kirkjubæjarklaustur appears to have had the task of supplying Skálholt cathedral with valuable textiles and offering educational training in that field. The excavation on the supposed site of the nunnery in Unartoq fjord likewise indicates an emphasis on textile making, perhaps to provide the cathedral in Garðar with churchly garments and textiles, in addition to food products acquired from the nunnery's properties. Similar sources for Reynistaðarklaustur are lacking, but it may be speculated that the nunnery there provided Hólar cathedral with supplies comparable to those provided by the two other nunneries mentioned here. However, there are some indications suggesting that the nuns in Reynistaðarklaustur also contributed to the ongoing production of books (Óskarsdóttir 2000). On the other hand, the monasteries appear generally to have focused on educating future priests and producing books, both liturgical and non-liturgical (Kristjánsdóttir 2017).

As may be expected, both written and archaeological sources show that the monasteries and nunneries in Iceland were staffed by people who dealt with the many and varied tasks of the monastic communities living there. Likewise, the excavation on the supposed site of the Benedictine nunnery in Uunartoq fjord in Norse Greenland points towards a large household, but further research on the Augustinian house operating in the country is lacking. Still, in both countries, Iceland and Norse Greenland, the monastic houses appear not to have competed with one other but rather to have shared the tasks of meeting the needs of the community, based on the skills and resources each possessed. Regardless, in light of the overall organisation of the Church in Iceland and Norse Greenland, the Benedictine nunneries and Augustinian monasteries seem to have been well suited to both societies, despite being located in the northernmost periphery of the Roman Catholic world.

**Funding:** This research was funded by University of Iceland Research Fund.

**Conflicts of Interest:** The author declares no conflict of interest.

## Notes

1. The so-called *Eigenkirckenwesen* in German and *privatkyrkosystem* in Swedish.
2. In Icelandic: *bænhús*.
3. A contemporary source, Flateyjarbók (1945, p. 241), also lists the churches in Norse Greenland, including the cathedral in Garðar, but the monastic houses are not mentioned explicitly there.
4. The original is published in *Diplomatarium Norwegium*, vol. 10, p. 15.

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
