# Peer review of "Medieval Monasticism in Iceland and Norse Greenland"

_religions, doi:10.3390/rel12060374_

Round 1

Reviewer 1 Report

This is a well done study on Iceland's and Greenland's monastic culture for international readers. Perhaps, there could be some more remarks about the dissolution of the houses.

Some orthographical and stylistic corrections:

Lines 228-31: Interestingly, on the same day, Bishop Árni in Bergen sent an almost identical letter to Bishop Árni Helgason in Skálholt for to gratitude him prayed for the soul of King Erik II and five recently deceased Norwegian bishops. > to be modified

Line 397: The ruin of a church and a surrounding cemetery was excavated > were

Lines 434-35; The large number of artifacts connected to the household, such as barrels, vessels, plates, spoons, and ladles, implies abundant food supplies > imply

There is the question whether, depending on the style of the journal, the Icelandic and Norwegian titles in the bibliography shouldn’t also be offered in English translation; and, generally, are the titles in the bibliography following  the style sheet of the journal?

Author Response

Thanks for the review. I have fixed all what was requested of reviewer 1. See attachment.

Reviewer 2 Report

It was a good article. I'm impressed by the clarity of arguments used, the consistency of the text, and its intellectual value.

Author Response

Thanks a lot.

Reviewer 3 Report

237 McClain 2012

284 churchly = a strange word

410 english = not clear

Author Response

Thanks for the review. I have fixed all what was required from reviewer 3. See attachment.
